# Effect of Cephalosporin Treatment on the Microbiota and Antibiotic Resistance Genes in Feces of Dairy Cows with Clinical Mastitis

**DOI:** 10.3390/antibiotics11010117

**Published:** 2022-01-17

**Authors:** Lei Dong, Lu Meng, Huimin Liu, Haoming Wu, Martine Schroyen, Nan Zheng, Jiaqi Wang

**Affiliations:** 1Key Laboratory of Quality & Safety Control for Milk and Dairy Products of Ministry of Agriculture and Rural Affairs, Institute of Animal Sciences, Chinese Academy of Agricultural Sciences, Beijing 100193, China; donglei@caas.cn (L.D.); menglu@caas.cn (L.M.); liuhuimin02@caas.cn (H.L.); wuhaoming@caas.cn (H.W.); 2Laboratory of Quality and Safety Risk Assessment for Dairy Products of Ministry of Agriculture and Rural Affairs, Institute of Animal Sciences, Chinese Academy of Agricultural Sciences, Beijing 100193, China; 3Precision Livestock and Nutrition Laboratory, Teaching and Research Centre (TERRA), Gembloux Agro-Bio Tech, University of Liège, 5030 Gembloux, Belgium; martine.schroyen@uliege.be

**Keywords:** dairy cow mastitis, cephalosporin, fecal microbiota, antibiotic resistance genes

## Abstract

Antibiotics are frequently used to treat dairy cows with mastitis. However, the potential effects of β-lactam antibiotics, such as cephalosporins, on the fecal microbiome is unknown. The objective was to investigate the effects of ceftiofur and cefquinome on the fecal microbiota and antibiotic resistance genes of dairy cows with mastitis. The fecal samples were collected from 8 dairy cows at the following periods: the start day (Day 0), medication (Days 1, 2, and 3), withdrawal (Days 4, 6, 7, and 8), and recovery (Days 9, 11, 13, and 15). 16S rRNA gene sequencing was applied to explore the changes in microbiota, and qPCR was used to investigate the antibiotic resistance genes. The cephalosporin treatment significantly decreased the microbial diversity and richness, indicated by the decreased Shannon and Chao 1 indexes, respectively (*p* < 0.05). The relative abundance of *Bacteroides*, *Bacteroidaceae*, *Bacteroidales*, and *Bacteroidia* increased, and the relative abundance of *Clostridia*, *Clostridiales*, *Ethanoligenens*, and *Clostridium* IV decreased at the withdrawal period. The cephalosporin treatment increased the relative abundance of β-lactam resistance genes (*bla*_TEM_ and *cfx*A) at the withdrawal period (*p* < 0.05). In conclusion, the cephalosporin treatment decreased the microbial diversity and richness at the medication period, and increased the relative abundance of two β-lactam resistance genes at the withdrawal period.

## 1. Introduction

Mastitis is regarded as one of the most frequent diseases in the dairy cows. Mastitis can directly and/or indirectly affect milk hygienic quality [1] and can lead to a substantial loss in milk production [2,3]. The average economic losses due to mastitis are estimated at around USD 325 per cow per year [4]. About 24% of antibiotics in the dairy industry are used for mastitis treatment and around 44% for mastitis prevention [5]. Antibiotic usage (both oral and injection) has a profound influence on the microbiome of animal feces, leading to an increase in the relative abundance of antibiotic resistance genes [6,7]. There is also a concern that due to the continued antibiotic use to prevent or treat mastitis, antimicrobial resistance will greatly increase the difficulty and cost of treatment [8]. It is reported that the chlortetracycline treatment promoted the abundance of tetracycline resistance genes such as *tet*(A) and *tet*(W) [6]. It was reported that 27% of oxytetracycline was excreted from the gastrointestinal tract when this antibiotic was injected into cattle, and therefore the microbiota in the feces were exposed to the antibiotic [7]. Antibiotic resistance genes (ARGs) can be transferred into the environment and pose a high risk to soil ecology and public health [9]. The abundance of ARGs in fertilized greenhouse soils was higher than that in field soil [10].

Cephalosporins are frequently used in veterinary medicine to treat bacterial infections [11]. Ceftiofur is a third-generation cephalosporin that was approved for veterinary use to treat various Gram-negative bacterial infections. Ceftiofur is also one of the most used antibiotics in dairy cows to treat mastitis, metritis, and respiratory diseases [12], and is the only third-generation cephalosporin approved for veterinary use in the USA [13]. Cefquinome is a fourth-generation cephalosporin developed solely for veterinary use. Cefquinome can treat the infections caused by *Staphylococcus aureus*, *Streptococcus suis* Serotype 2, and *Escherichia coli* [14,15,16]. It was reported that bacteria in animals can develop resistance to cefquinome [17,18]. A Swiss study found up to 44% of *Escherichia coli* isolates resistant to cefquinome [19]. There is a reasonable concern that the use of cefquinome in cows could increase the expression of β-lactam genes, which may deliver resistance to cephalosporins [11]. Resistance to cephalosporin was related to the production of β-lactamases [20]. It is reported that β-lactamase encoding genes, such as *bla*_TEM_, *bla*_CMY_, *bla*_SHV_, and *cfxA*, confer resistance to ceftiofur [11,21]. To date, no research has investigated the effect of cephalosporins on the fecal microbiome of mastitis cows.

The objective was to investigate the alterations in fecal microbiota and the antibiotic resistance genes following cephalosporin treatment of mastitis cows.

## 2. Results

### 2.1. Animal Weight Gain, and 16S rRNA Gene Sequencing Overview

None of the dairy cows in this research received any other antibiotic treatments. The dairy cows were weighed prior to the medication and the recovery period. The growth rate of the dairy cows was not influenced by the antibiotic medication (*p* > 0.05). The raw reads of sequences per sample ranged from 35,133 to 105,031. After cleaning, the tags of sequences per sample ranged from 33,984 to 102,801. The OTU numbers ranged from 795 to 2344.

### 2.2. α Diversity

The rarefaction analysis performed for each fecal sequence dataset retrieved rarefaction curves. The result suggested that the sample size was large enough to represent the bacterial diversity present in the communities (Figure 1). Microbial diversity within the α diversity was measured by richness (Chao 1) and diversity indices (Shannon and Simpson). The average values of the Shannon diversity index and the average Chao 1 index for the microbiota were similar among the different periods, which both decreased significantly at the medication period (*p* < 0.05), suggesting that the total number of species and the abundance of microbiota were all decreased. However, the mean values of the Simpson index did not show significant differences among the different periods (Figure 2).

### 2.3. Microbial Community Analysis

The distribution of the most abundant classes and families in the fecal samples are displayed in Appendix A. In the feces, bacteria are the main microbes. At the class level, Clostridia was the most abundant bacterium at all four periods, accounting for about 60% of all bacteria, followed by *Bacteroidia*, *Actinobacteria*, and *Erysipelotrichia*. These four classes accounted for over 85% of all bacteria (Appendix A). At the family level, a total of 50 taxa were detected in the feces. *Ruminococcaceae*, *Lachnospiraceae*, *B**ifidobacteriaceae*, and *Porphyromonadaceae* were the four most abundant (Appendix A).

To obtain further insights, the statistical differences in genera in the fecal samples were analyzed using the Vegan R package. At the genus level, *Bifidobacterium*, *Sporobacter, Bacteroides*, *Clostridium* sensu stricto, *Romboutsia*, and *Ruminococcus* dominated the fecal samples (Figure 3a). *Bifidobacterium* belongs to *Actinobacteria*; *Sporobacter*, *Clostridium* sensu stricto, *Romboutsia*, and *Ruminococcus* belong to *Firmcutes*; and *Bacteroides* belongs to *Bacteroidetes* (Figure 3a). *Roseburia* was more abundant when antibiotics were used. The relative abundance of *Bacteroides*, *Bacteroidaceae*, *Bacteroidales*, and *Bacteroidia* increased with the cephalosporin treatment, and the relative abundance of *Clostridia*, *Clostridiales*, *Ethanoligenens*, and *Clostridium_IV* decreased (Figure 3b). At the recovery period, the relative abundance of *Clostridium XI*, *Peptostreptococcaceae*, *Verrucomicrobiales*, *Verrucomicobiae*, *Verrucomicrobia*, *Akkermansia*, and *Verrucomicrobiaceae* increased in the fecal samples (Figure 3b). Appendix A displays that the antibiotics were effective in reducing or preventing the growth of *Moraxellaceae*.

### 2.4. Antibiotic Resistance Genes

The proportions of sixteen antibiotic resistance genes (*cfx*A, *bla*_ROB_, *bla*_CMY_, *mec*A, *bla*_CTX-M_, *bla*_1_, *bla*_TEM_, *str*A, *str*B, *erm*(A), *erm*(B), *sul*1, *sul*2, *tet*(A), *tet*(B)*, tet*(C), *tet*(H), *tet*(Q), *van*C and *van*G) in fecal samples from four different sampling periods were quantified. Among these 20 resistance genes, only eight genes (*cfx*A, *bla*_TEM_, *bla*_CMY_, *str*B, *tet*(A), *tet*(B), *tet*(C) and *tet*(Q)) showed Cq values during detection in fecal samples. Ceftiofur and cefquinome significantly increased the proportion of *bla*_TEM_ and *cfx*A in the feces at the withdrawal period when compared with Day 0 (Figure 4). Other resistance genes were not affected by the antibiotic treatment (data not shown). At the recovery period, the proportion of *cfx*A was significantly decreased compared with the withdrawal period, but the proportion of *bla*_TEM_ was not changed significantly.

## 3. Discussion

Antibiotics such as piperacillin/tazobactam (TZP) or ceftriaxone (CRO) have significantly decreased the diversity of the microbiome in feces of mice, and dysbiosis was more obvious and prolonged after five days of CRO exposure [22]. These studies supported our results that antibiotics can decrease the diversity in feces at the medication period.

It was reported that the relative abundance of class *Bacteroidia* was increased and the relative abundance of class *Actinobacteria* was decreased in experimental cattle after 3 days of ceftiofur exposure [11]. It was similar to our study that the relative abundance of *Bacteroidia* was increased in feces at the medication period. It was also reported that the *Bacteroides* strains isolated from human infections contained β-lactamase genes, which reduced sensitivity to antibiotics [23]. Therefore, we speculated that the increase in *Bacteroidia* at the medication period may be related to the increase of β-lactam resistance genes.

The *bla*_TEM_ has been reported to have a high prevalence in the ceftiofur-resistant bacteria of swine tissues [24]. In accordance with our previous study, the *bla*_TEM_ in milk that were collected from the cows was significantly increased (*p* < 0.05) after 3 days of cephalosporin exposure [25]. It was also reported that *bla*_TEM_ was one of the most abundant ARGs in the feces from a pig farm [26]. In the present study, the β-lactamase encoding genes *bla*_TEM_ and *cfx*A were significantly increased (*p* < 0.05) at the withdrawal period. *cfx*A was considered to be the most abundant β-lactam ARG in ceftiofur-treated cattle feces [11]. *cfx*A was also found to be an important gene encoding β-lactamase in *Bacteroides* spp. [27]. Avelar et al. [28] detected the *cfx*A gene in 11 *Bacteroides* spp. strains among a total of 73 strains. So, there may be a close correlation between the *Bacteroides* in the feces and the increased genes encoding β-lactam.

The milk microbiota richness of mastitis cows treated with cephalosporins did not decrease [25]. In the present study, we investigated the effect of cephalosporins on the feces of cows. The results suggested that the cephalosporin treatment indeed affected the abundance of the microbiota in feces, with a decreased richness and decreased diversity, suggesting that antibiotics may have a more pronounced effect on the gut than milk. It may be because the ceftiofur sodium has a pKa value of 3.7 and insufficient lipid-soluble properties to penetrate breast milk [29]. It was reported that when the cattle were treated with ceftiofur, the β-lactam ARGs in feces were increased, and the ceftiofur-resistant *E. coli* isolates from the feces were greater compared to control cattle [30]. In a recent study that used qPCR to detect β-lactam ARGs, they were found to be increased in the feces of cows treated with ceftiofur compared to cows without ceftiofur treatment [11]. Considering the proportion of *bla*_TEM_ and *cfx*A in feces during the withdrawal period in this study, it seems that the feces might be an important reservoir of the *bla*_TEM_ and *cfx*A genes. Other studies have also confirmed that animal feces are an important reservoir of ARGs [23].

Although none of the cattle received tetracyclines during the period of the study, some antibiotic resistance genes coding for tetracycline resistance (*tet*(W) and *tet*(Q)) were detected but did not significantly change at the different periods (data not shown). This may be because the treatment of a kind of antibiotic can provide selective pressure to maintain other unassociated resistance genes by linking to mobile genetic elements (MGEs) [31]. The co-transfer of *erm*(B) and *tet*(M) in the presence of erythromycin has been reported in *Streptococcus pyogenes* isolates [32], and the colocalization of *mef*A, *aph*A3, *tet*(Q), and IS614 was observed in a transposon of Bacteroide [33]. It has been reported that MGEs promote the mobilization and spread of ARGs in bacteria. High concentrations of ARGs are considered a risk to public health because the ARGs can transfer from the manure compost, becoming pathogens in agricultural soil [23]. The resistant bacteria and resistance genes in the feces can also be seen as a serious problem because they may transfer among cattle and result in antibiotic treatment failure. Therefore, the appropriate use of antibiotics in dairy cattle is an important process to avoid the spread of ARBs and ARGs.

## 4. Materials and Methods

### 4.1. Animals and Sample Collections

Fecal samples were collected from a dairy farm in Tianjin city, China. The dairy cows were judged to suffer from clinical mastitis by a veterinarian based on the obvious symptoms of redness of either udder, milk curdling, discoloration, and when the somatic cell count in milk was more than 500,000 cells/mL. The somatic cell count was calculated by the California mastitis test (CMT). A total of 8 primiparous mastitis-affected Holstein dairy cows (one quarter was infected; 560–686 kg body weight; 105–226 days in milk; 34.26–39.12 kg of milk per day) were selected. The cows had not been treated with any antibiotics prior to the study. According to the uniform regulations of the dairy farm, these mastitis cows were housed individually in a well-ventilated barn and fed a totally mixed ration three times daily. Prior to antibiotic treatment, feces from these cows were collected from the rectum by the veterinarian, grabbed with sterile groves. Approximately 300 g of feces collected for the first time were discarded to prevent contamination. Then 100 g of the fresh samples were immediately placed into a 200 mL-sterile plastic bottle to avoid exposure to the environment. The cows were then injected with ceftiofur sodium for injection (Qilu animal health products co., LTD, Shandong, China) into the muscle (2 mg/kg body weight), and with a cefquinome sulfate intermammary infusion (Qilu animal health products co., LTD, Shandong, China) into the teat canal of the mastitis-affected quarter (0.75 ng/kg body weight) by the veterinarian once per day from Day 1 to Day 3. The somatic cell counts in milk returned to normal values (<200 thousand cells/mL), and the dairy cows stopped receiving the antibiotics after Day 3. The milk was tested by Delvotest SP-NT (DSM Food Specialities R&D, Delft, The Netherlands) according the manufacturer’s instructions to ensure that the antibiotics were not detected in the milk on Day 9. The feces of all dairy cows were then sampled at Days 1, 2, 3, 4, 6, 8, 9, 11, 13, and 15 (Figure 5). Day 0 referred to the start day; Days 1 to 3 were classified as the medication period; Days 4 to 8 were classified as the withdrawal period; and Days 9 to 15 were classified as the recovery period.

### 4.2. DNA Extraction

Total DNA was extracted from 500 mg of each fecal sample using the E.Z.N.A™ Mag-Bind Soil DNA Kit (OMEGA, Norcross, GA, USA), according to manufacturer’s instructions. The DNA samples’ quality and concentration were measured using a Qubit 3.0 DNA detection kit (Life Technologies, Grand Island, NY, USA). These DNA samples were stored at −80 °C for further genotypic quantification.

### 4.3. PCR Amplification

The V3–V4 region of the 16S rDNA was amplified by PCR (94 °C for 3 min, 94 °C for 30 s, 45 °C for 20 s, and 65 °C for 30 s for 5 cycles). Illumina bridge PCR compatible primers were introduced in the second round of PCR amplification at 94 °C for 20 s, 55 °C for 20 s, and 72 °C for 30 s for 20 cycles, and finally extended at 72 °C for 5 min using primers 341F: CCTACGGGNGGCWGCAG; 805R: GACTACHVGGGTATCTAATCC [34]. Triplicate PCR reactions were performed with 30 μL of the mixture containing 15 μL of 2 × Hieff^®^ Robust PCR Master Mix, 1 μL of Primer F, 1 μL of primer R, 10–20 ng template DNA or PCR products, and H_2_O was added to 30 μL. All PCR reagents were from TOYOBO, Japan.

### 4.4. Illumina Novaseq 6000 Sequencing

The amplicons were extracted from a 2% agarose gels and purified using the SanPrep DNA Gel Extraction Kit (SANGON Biotechnology, Shanghai, China), according to the manufacturer’s instructions, and quantified using the ABI Step One Plus Real-Time PCR System (Life Technologies, Foster City, CA, USA). Purified amplicons were concentrated in equimolar and paired-end sequenced (PE250) on the Illumina platform according to the standard protocols.

### 4.5. Quantification of Antibiotic Resistance Genes

The quantity of 20 antibiotic resistance genes was evaluated using qPCR, as described previously [7]. In brief, genes conferring resistance to beta-lactams (*cfx*A, *bla*_ROB_, *bla*_CMY_, *mec*A, *bla*_CTX-M_, *bla*_1_, and *bla*_TEM_), aminoglycosides (*str*A and *str*B), macrolides (*erm*(A) and *erm*(B)), sulfonamides (*sul*1 and *sul*2), tetracyclines (*tet*(A), *tet*(B), *tet*(C), *tet*(H) and *tet*(Q)), and vancomycin (*van*C and *van*G) were evaluated. The primer sequences used were as previously described in Huang et al. [35]. These genes were normalized against the 16S rRNA gene, which was also quantified by qPCR. The 16S rRNA gene was amplified using the 357-F: 5′-CCTACGGGAGGCAGCAG-3′ and 518-R: 5′-ATTACCGCGGCTGCTGG-3′ primers that were also used to generate the 16S rRNA gene libraries.

### 4.6. Statistical Analysis

Raw data including adapters or low-quality reads would affect the assembly and following analysis. Therefore, in order to obtain high quality clean reads, FASTP [36] (version 0.18.0) was used to further filter the original reads according to the following rules. The UPARSE [37] (version 9.2.64) pipeline was used to cluster the effective tags into operational taxonomic units (OTUs) with similarity ≥ 97 %. This package was also used to calculate the Jaccard and Bray–Curtis distance matrix. The α-diversity indexes, such as Chao 1, Shannon, and Simpson, were calculated in QIME [38] (version1.9.1). The rarefaction analysis was performed using the mothur [39]. The alpha index comparison among groups was calculated by Tukey’s HSD test and the Kruskal–Wallis H test using the Vegan package in R [40]. Linear discriminant analysis effect size (LEfSe) was used to determine which microorganisms were significantly different among groups [41]. The antibiotic resistance gene comparison among groups was statistically analyzed using ANOVA with Tukey’s multiple comparison test by SPSS software version 24.0 (SPSS, Inc., Chicago, IL, USA).

## 5. Conclusions

This study provides a snapshot of the changes in the fecal microbiota and resistome affected by cephalosporins. The richness and diversity of the bacterial communities were significantly decreased at the medication period. The relative abundance of *Bacteroides*, *Bacteroidaceae*, *Bacteroidales*, and *Bacteroidia* increased, and the relative abundance of *Clostridia*, *Clostridiales*, *Ethanoligenens*, and *Clostridium* IV decreased at the withdrawal period. This research suggests that cephalosporins had a measurable and immediate effect on the fecal microbiota. However, the cephalosporins increased the proportion of the β-lactam genes *bla*_TEM_ and *cfx*A at the withdrawal period. The long-term (>10 days) effect of the cephalosporin treatment on the fecal microbiota and resistome are worthy of further investigation. At the same time, it is important to develop the appropriate management to control the transfer of ARGs.

## Figures and Tables

**Figure 1 antibiotics-11-00117-f001:**
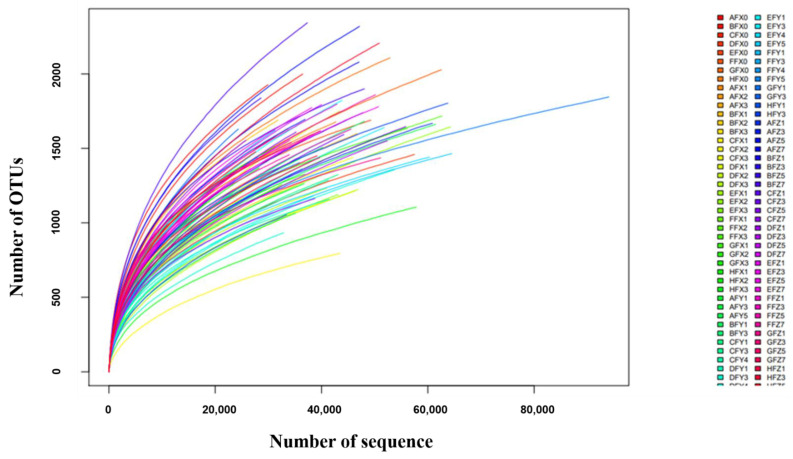
The rarefaction analysis performed using mothur. The rarefaction curve based on the species diversity showed sufficient coverage for the sequences. The first letter of the number (A–H) represented the different cows. X0, X (X1, X2, X3), Y (Y1, Y3, Y4, Y5), and Z (Z1, Z3, Z5, Z7) indicate the periods Day 0, medication, withdrawal, and recovery, respectively.

**Figure 2 antibiotics-11-00117-f002:**
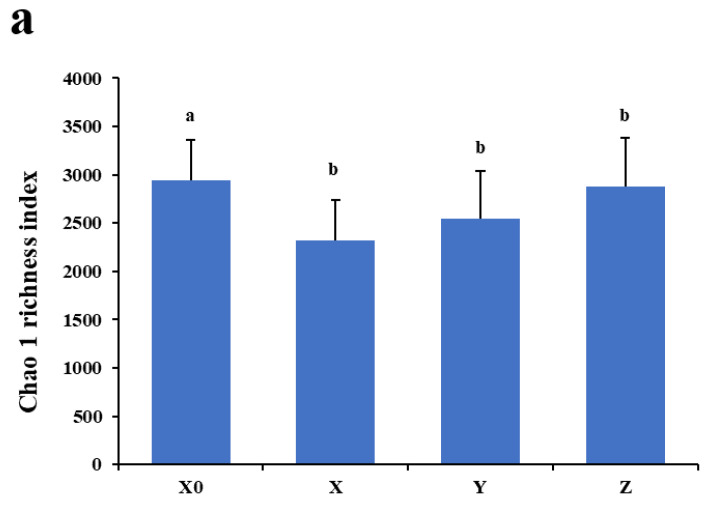
(**a**) Chao 1 richness index of samples from four different periods. (**b**) Shannon diversity index of samples from four different periods. (**c**) Simpson diversity index of samples from four different periods. Different lowercase letters within each sampling group represent significantly different means (*p* < 0.05). X0, X, Y, and Z indicate the periods Day 0, medication, withdrawal, and recovery, respectively.

**Figure 3 antibiotics-11-00117-f003:**
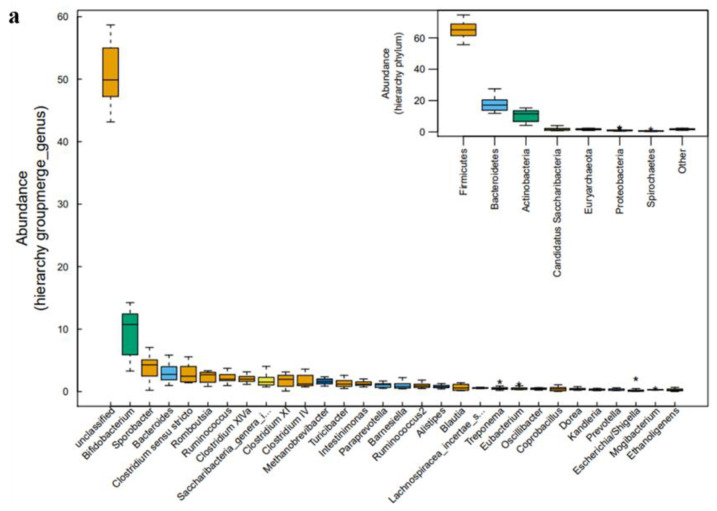
Alterations in bacterial genera abundance in the feces at different periods. (**a**) Selected genera belonging to a different phylum with significantly different abundances in the feces. (**b**) Lefse analysis histogram; different colors represent different groups. The horizontal axis is the LDA score obtained after LDA analysis, and the vertical axis is the group of microorganisms with significant effects.

**Figure 4 antibiotics-11-00117-f004:**
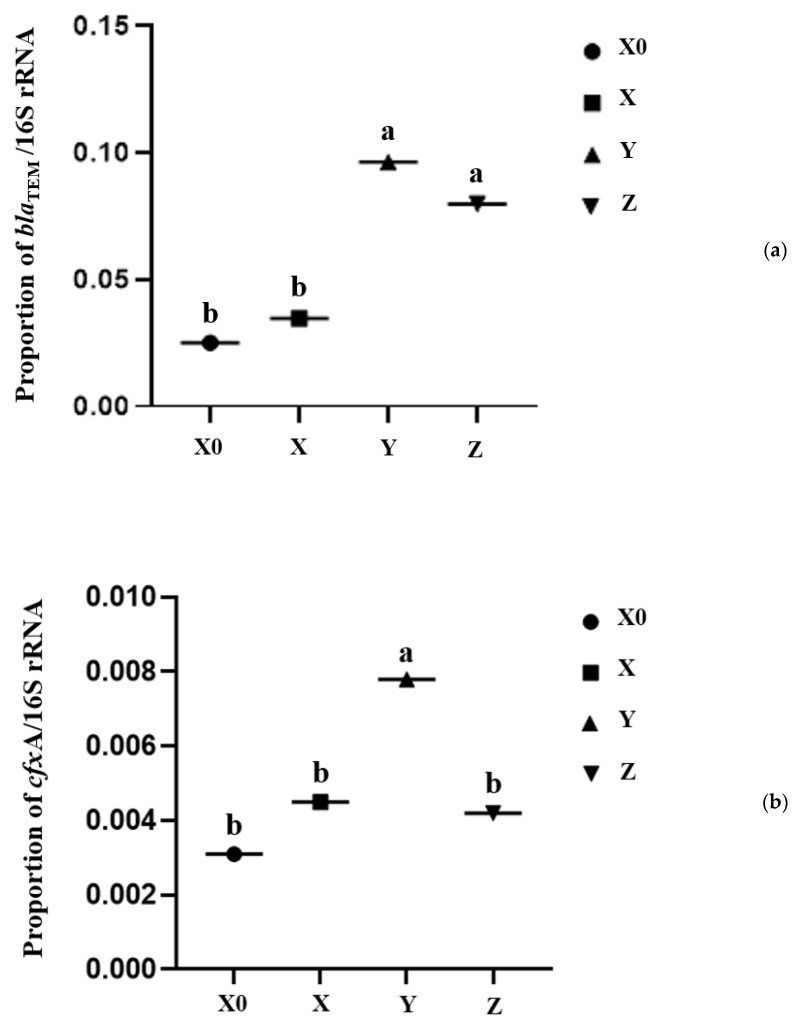
Proportion of the antibiotic resistance genes (**a**) *bla*_TEM_ and (**b**) *cfx*A compared with the 16S rRNA gene. Different lowercase letters suggest significantly different means (*p* < 0.05). X0, X, Y, and Z indicate the periods Day 0, medication, withdrawal, and recovery, respectively.

**Figure 5 antibiotics-11-00117-f005:**
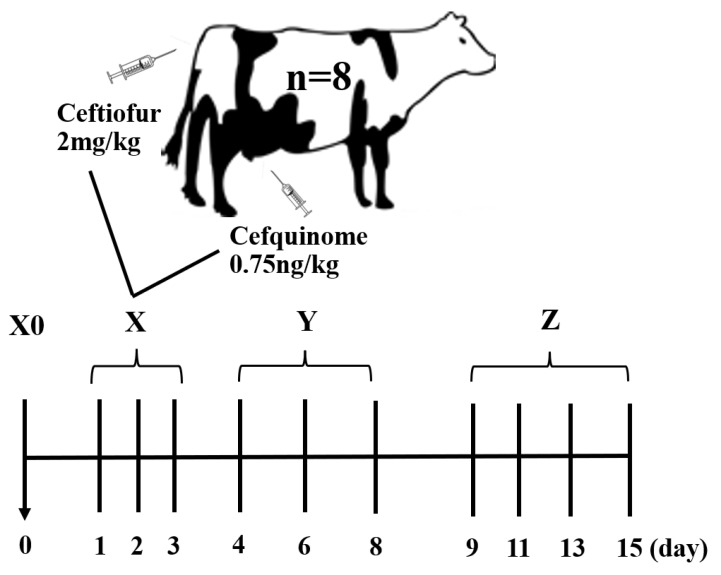
Timeline for fecal sampling. Sampling days are displayed above the black lines. The antibiotic treatments are noted at Days 1, 2, and 3. X0, X, Y, and Z indicate the periods Day 0, medication, withdrawal, and recovery, respectively.

## Data Availability

All 16S rRNA gene sequences were submitted in the Sequence Read Archive (SRA) under BioProject accession number PRJNA758197.

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
