# Peer review of "Effect of Cephalosporin Treatment on the Microbiota and Antibiotic Resistance Genes in Feces of Dairy Cows with Clinical Mastitis"

_antibiotics, 2022, doi:10.3390/antibiotics11010117_

Round 1

Reviewer 1 Report

It is usually suggested that material and methods part is before results part.

Special comments

L50-53: repeated sentences

L202-203: improvement in English language

L199-200: The mastitis definition is not only producing high number of somatic cells, nor uniquely one criterion. Some additional information about the clinical sighs could be added. It is not usual practice to use antibiotics especially cephalosporin, only with elevated high somatic cells in milk. Additionally the method of cell count could be added here. Moreover, mastitis in one quarter per cow? Did microbiological milk cultures performed from affected quarters? It is not clear

L207: sentence needs clarification

L209-211: Injectable intramuscular solution should be added (name, manufacturer). It is not clear how was the intramammary administration performed. Was a commercial intramammary solution used? Was a special cefquinome solution made by the researchers? Was intramammary administration to 1 or more mammary quarters in each cow? It should be added in the text.

L211: add once per day

L212: Two lines above it is mentioned that antibiotic administration was done at day 1, 2 and 3. Here it is mentioned till day 4, It is needs language improvement as not to be misleading here in text (it is very clear in Figure 4)

Reviewer 2 Report

The manuscript entitled: “Effect of cephalosporin treatment on the microbiota and antibiotic resistance genes in feces of dairy cows with mastitis” is a well designed study focusing outcome of specific antibiotics on AMR genes and microbial community in fecal materials originated from bovine mastitis. Considering the bovine mastitis which is a major production disease of dairy cattle globally and AMR- a global health crisis and pandemic, it’s a time demanding study is of significance interest in the aligned field.

My comments are as follows:

Line 72. Use the term “gene” after “16S rRNA” and follow this in other cases where applicable. There is no such term as “16S rRNA” alone.

Line 79. Please show the rarefaction cure to show data presented here has enough  coverages of diversity.

Line 131-132: what was the detection limit, please mention  here.

Line 147.please use “five” not “5”, usually we write 1 to 9 in words and 10 to beyond in number..??

Line 150. Is the any report or evidence that there are less or no ceftiofur /cephalosporing gene in Actinobacteria, since in this study ceftiofur was decreased in experimental cattle after 3  days of ceftiofur exposure ??

Line 160. What environment you are refereeing here, please mention it…

Line 167.  Which sample of swine?? Feces…Please mention it here??

Lime 172. Any idea  why the antibiotics cephalosporin may have a more pronounced effect on the gut than  milk.  Pharmacodynamics??

Line 187. Make Streptococcus pyogenes Italic..all bacterial name has to be italic

Line 198. What was the weight of the fecal samples collected from each cattle?? Please make it clear…

Line 200. Are you talking about subclinical mastitis or clinical mastitis/somatic cell counts are usually done for subclinical mastitis, so, please make it clear: subclinical or clinical mastitis and also MAKE IT CLEAR in the title of the manuscript (what type of mastitis??)

Line 201. The number of animals sampled was only 8, is it a good number, any data to support that n=8 is enough for this type statistical analysis used here??

Line 250. There are several other genes responsible for resistance to tetracycline like tetA and vancomycin like vanA, and why they were not used in this study, any explanation??

Line 255. 16S not 16s
